Publication rate and citation counts for preprints released during the COVID-19 pandemic: the good, the bad and the ugly

http://orcid.org/0000-0003-1829-7001 Añazco Diego 1
http://orcid.org/0000-0002-7043-5515 Nicolalde Bryan 1
Espinosa Isabel 1
Camacho Jose 2
Mushtaq Mariam 1
Gimenez Jimena 1
http://orcid.org/0000-0001-6979-5655 Teran Enrique 1 eteran@usfq.edu.ec
1 Colegio de Ciencias de la Salud, Universidad San Francisco de Quito , Quito , Ecuador
2 Colegio de Ciencias e Ingenieria, Universidad San Francisco de Quito , Quito , Ecuador
Gray Andrew
Electronic publication date: 2021 Mar 3
Publication date: 2021
Volume: 9
Electronic Location ID: e10927
Received 2020 Sep 25; Accepted 2021 Jan 20
Copyright: © 2021 Añazco et al.
Copyright year: 2021
Copyright holder: Añazco et al.
License: This is an open access article distributed under the terms of the Creative Commons Attribution License, which permits unrestricted use, distribution, reproduction and adaptation in any medium and for any purpose provided that it is properly attributed. For attribution, the original author(s), title, publication source (PeerJ) and either DOI or URL of the article must be cited.
License URL: https://creativecommons.org/licenses/by/4.0/

Keywords: Preprints, COVID-19, Preprints servers, Crossref, iSearch COVID-19 portfolio

Funding: The authors received no funding for this work.

==============================
Background

Preprints are preliminary reports that have not been peer-reviewed. In December 2019, a novel coronavirus appeared in China, and since then, scientific production, including preprints, has drastically increased. In this study, we intend to evaluate how often preprints about COVID-19 were published in scholarly journals and cited.

Methods

We searched the iSearch COVID-19 portfolio to identify all preprints related to COVID-19 posted on bioRxiv, medRxiv, and Research Square from January 1, 2020, to May 31, 2020. We used a custom-designed program to obtain metadata using the Crossref public API. After that, we determined the publication rate and made comparisons based on citation counts using non-parametric methods. Also, we compared the publication rate, citation counts, and time interval from posting on a preprint server to publication in a scholarly journal among the three different preprint servers.

Results

Our sample included 5,061 preprints, out of which 288 were published in scholarly journals and 4,773 remained unpublished (publication rate of 5.7%). We found that articles published in scholarly journals had a significantly higher total citation count than unpublished preprints within our sample (p < 0.001), and that preprints that were eventually published had a higher citation count as preprints when compared to unpublished preprints (p < 0.001). As well, we found that published preprints had a significantly higher citation count after publication in a scholarly journal compared to as a preprint (p < 0.001). Our results also show that medRxiv had the highest publication rate, while bioRxiv had the highest citation count and shortest time interval from posting on a preprint server to publication in a scholarly journal.

Conclusions

We found a remarkably low publication rate for preprints within our sample, despite accelerated time to publication by multiple scholarly journals. These findings could be partially attributed to the unprecedented surge in scientific production observed during the COVID-19 pandemic, which might saturate reviewing and editing processes in scholarly journals. However, our findings show that preprints had a significantly lower scientific impact, which might suggest that some preprints have lower quality and will not be able to endure peer-reviewing processes to be published in a peer-reviewed journal.

Introduction

A preprint is a preliminary report of scientific work that is shared publicly before it has been peer-reviewed. Most preprints have a digital object identifier (DOI) and can be cited by other research articles (Kaiser, 2019). There are multiple servers that host preprints related to biological and medical sciences, such as bioRxiv (https://www.biorxiv.org/) and medRxiv (https://www.medrxiv.org/), operated by Cold Spring Harbor Laboratory; and Research Square, another widely used platform during the COVID-19 pandemic (https://www.researchsquare.com/). The validity and usage of information from preprints are being discussed. Advocates claim that preprints accelerate access to findings and improve the quality of published papers by permitting faster feedback before publication. Also, the audience for preprints might be larger because many scholarly journals do not have open access (Kaiser, 2017). Those who are against the use of preprints state that many of these investigations may be flawed due to the lack of a peer-review process (Chalmers & Glasziou, 2016). Thus, readers should be aware that preprints might contain errors and that information reported there has not been peer-reviewed or endorsed by the scientific or medical community (Brainard, 2020; Kaiser, 2017, 2019).

Coronavirus disease 2019 (COVID-19) was first described in December 2019 in Wuhan, China, and the World Health Organization (WHO) declared it as a pandemic on March 12, 2020 (WHO, 2020). Due to the drastic increase in scientific production about this novel disease, scholarly journals have been overwhelmed, and in response, some journals have accelerated their usual processes to shorten the time it takes to publish an article (Brainard, 2020). Nonetheless, due to the imperative need for rapid information, authors have been urged to share relevant information as soon as possible; thus, an increasing number of studies have been posted as preprints (Rubin et al., 2020). The International Committee of Medical Journal Editors (ICMJE) stated that in the case of public health emergencies, information with immediate implications for public health should be rapidly disseminated, without concerns that this will prevent subsequent publication in a scholarly journal (ICMJE, 2020). The World Health Organization and many scholarly journals require that authors share their work initially using a preprint server before it goes through formal peer-reviewing and editing processes (Fidahic et al., 2020; Moorthy et al., 2020).

Thus, the objective of this study was to determine the publication rate in scholarly journals for preprints about COVID-19 posted from January 1, 2020 to May 31, 2020, in medRxiv, bioRxiv, and Research Square. Also, we evaluated how often preprints or published papers were cited, and compared preprint servers in terms of publication rate, citation count, and time to publication.

Materials and Methods

Search strategy

We used the iSearch COVID-19 portfolio (https://icite.od.nih.gov/covid19/search/), a comprehensive platform developed by the National Institute of Health (NIH) that hosts publications on scholarly journals and preprints related to SARS-CoV-2 and COVID-19, to obtain our sample of preprints. We searched on August 10, 2020, to identify all the preprints that had been posted on three different preprint servers (medRxiv, bioRxiv, and Research Square) from January 1, 2020 to May 31, 2020 (Fig. 1). The search yielded 5,106 preprints, and after excluding 45 duplicates, we had a sample of 5,061 preprints. For duplicated preprints, we decided to use the version that was first uploaded to a preprint server. We gathered metadata for these preprints, including the title, authors list, Digital Object Identifier (DOI), preprint server, and date of posting on the preprint server.

Figure 1 Timeline of data gathering for the study of preprints on COVID-19.

Publication status and citation count

If a preprint is subsequently published in a scholarly journal, a link to the published version is made available on the preprint server. We used Crossref (https://www.crossref.org/) to determine the publication status and citation count of this sample. To gather specific metadata from the Crossref application programing interface (API), we developed a program using Python (Python Software Foundation, https://www.python.org/). The codes for this program are available online (https://github.com/josecxz/Crossref-Adquisition). Using this method, we retrieved validated data for citation counts and the DOI of the published version of a preprint, if available. We could not gather data for 13 preprints using the API in Crossref, so a manual search was performed on the respective preprint server, Google Scholar, and PubMed to determine the publication status. However, we decided not to include citation counts for this set of preprints, as this data was not validated and methods to assess citation counts might differ. Also, duplicated preprints were removed before this search, and this data was not collected nor considered for subsequent analyses. After identifying preprints that were published in a scholarly journal, we searched Crossref using our program to determine the citation count for the published versions and the date in which these articles were published in a journal. Both searches on the Crossref API were conducted on August 11, 2020.

Statistical analysis

After gathering our data, we observed that the number of citations varied significantly. Through graphical exploration, we identified that our data had multiple outliers. In our dataset, the median represented the center of the distribution more accurately than the mean, which is more sensitive to extreme values. As well, we conducted a Shapiro-Wilk test that showed that our data had a non-normal distribution (p < 0.001). Hence, we decided to use non-parametric tests for our statistical analyses regarding citation counts. We conducted Mann–Whitney U tests for our citation counts analyses for independent samples, and we conducted a Wilcoxon signed-ranked test for a paired analysis of citation counts before and after publication in a scholarly journal, as described in the Results section.

We used a chi-squared test to compare publication rates for preprints on three different preprint servers (bioRxiv, medRxiv, Research Square). The non-parametric Kruskal–Wallis test, followed by a posthoc Dunn’s test for pairwise comparisons, was used to compare citation counts and the time intervals from posting on a preprint server to publication in a scholarly journal.

We used a two-sided alpha value of 0.05 for our statistical analyses, except for the chi-squared tests (one-sided alpha value of 0.05). We used the Bonferroni correction method to adjust probability (p-adj) values in our posthoc analyses. All analyses were conducted on IBM SPSS Statistics for Windows, version 25 (IBM Corp., Armonk, NY, USA).

Results

We included 5,061 preprints from three preprint servers. In our sample, most of the preprints were obtained from medRxiv (n = 3,258), followed by Research Square (n = 1,011) and bioRxiv (n = 792). By August 11, 2020, only 288 preprints (5.7%) had been published in a scholarly journal. The median time interval from the date of posting on a preprint server to the date of our search was 110 days (min 72–max 205 days).

Citation counts

As mentioned before, citation count data was not available for 13 preprints (8 unpublished preprints, 5 published preprints), so this set of preprints was excluded for the following analyses. Citation counts ranged from 0 to 168 citations for preprints, while it ranged from 0 to 3,587 citations for published articles. Unpublished preprints (n = 4,765) had a mean citation count of 1.99 (median: 0; Q1–Q3: 0–1), while published articles (n = 283) had a total mean citation count of 72.12 (median: 9; Q1–Q3: 2–38). After conducting a Mann-Whitney U test, we determined that published articles had a significantly higher total number of citations than unpublished preprints (p < 0.001) (Table 1).

Table 1 Characteristics and analyses of citation counts data made for preprints posted on medRxiv, bioRxiv, and research square from January 1 to May 31, 2020.

			p-value	
	Unpublished preprints (n = 4,765)	Published preprints (n = 283) (total citation count)		
Mean	1.99	72.12	p < 0.001	
Range	168	3,757		
Minimum	0	0		
Maximum	168	3,757		
Median (Q1–Q3)	0 (0–1)	9 (2–38)		
	Published preprints (n = 283) (citation count of preprint version)	Published preprints (n = 283) (citation count of published version)		
Mean	6.29	65.83	p < 0.001	
Range	170	3,587		
Minimum	0	0		
Maximum	170	3,587		
Median (Q1–Q3)	1 (0–5)	7 (1–30)		
	Unpublished preprints (n = 4,765)	Published preprints (n = 283) (citation count of preprint version)		
Mean	1.99	6.29	p < 0.001	
Range	168	170		
Minimum	0	0		
Maximum	168	170		
Median (Q1–Q3)	0 (0–1)	1 (0–5)		
Note:

Q1, quartile 1; Q3, quartile 3.

For the set of preprints that were published at some point (n = 283), we compared the number of citations that these articles had before and after being published in a scholarly journal. The mean number of citations for these articles before being published (preprint status) was 6.29 (median: 1; Q1–Q3: 0-5). By using a Wilcoxon signed-ranked test, we identified that the citation count for the same set of articles was significantly higher after being published in a scholarly journal (mean: 65.8; median: 7; Q1–Q3: 1–30) (p < 0.001).

As well, we compared the citation counts of preprints that eventually got published (n = 283) to preprints that remained unpublished (n = 4,765). After conducting a Mann–Whitney U test, we found that published preprints had a higher number of citations as preprints (mean: 6.29; median: 1; Q1–Q3: 0–5) than preprints that remained unpublished (mean: 1.99; median: 0; Q1–Q3: 0–1) (p < 0.001). These results suggest that preprints that were published eventually had a higher impact in scientific production, even before they were published in a scholarly journal, than preprints that remained unpublished.

In our sample, 3,140 preprints (62%) had no citations at all. Within the set of preprints that remained unpublished, 3,047 preprints (64%) had no citations; while 93 preprints (32.9%) that were eventually published had no citations as preprints. We conducted the same statistical analyses previously described after excluding from preprints that eventually got published (n = 283) those without citations before (n = 93) plus those with no citations once published (n = 50). However, 34 of these 50 were already included in the 93 with no citations as preprints, and so only an additional 16 were removed for a final sample of 174 published preprints with at least one citation. The statistical significance was not altered (Table 2).

Table 2 Characteristics and analyses of citation counts data made for preprints posted on medRxiv, bioRxiv, and research square from January 1 to May 31, 2020 after excluding preprints without citations.

			p-value	
	Unpublished preprints (n = 1,718)	Published preprints (n = 174) (citation count of published version)		
Mean	5.52	104.58	p < 0.001	
Range	167	3,586		
Minimum	1	1		
Maximum	168	3,587		
Median (Q1–Q3)	2 (1–5.75)	20.50 (6–72.50)		
	Published preprints (n = 174) (citation count of preprint version)	Published preprints (n = 174) (citation count of published version)		
Mean	10.09	104.58	p < 0.001	
Range	169	3,586		
Minimum	1	1		
Maximum	170	3,587		
Median (Q1–Q3)	4 (2–9)	20.50 (6–70.25)		
	Unpublished preprints (n = 1,718)	Published preprints (n = 174) (citation count of preprint version)		
Mean	5.52	10.09	p < 0.001	
Range	167	169		
Minimum	1	1		
Maximum	168	170		
Median (Q1–Q3)	2 (1–5.75)	4 (2–9)		
Note:

Q1, quartile 1; Q3, quartile 3.

Comparing preprint servers

Preprints in medRxiv had the highest number of published preprints and the highest publication rate (n = 225; 6.9%), compared to preprints in bioRxiv (n = 45; 5.7%) and Research Square (n = 18; 1.8%). We performed a chi-squared test that confirmed a significant difference regarding publication status on these databases (df = 2; χ2 = 37.77; p < 0.001). Post-hoc pairwise comparisons revealed significant differences when comparing bioRxiv to Research Square (df = 1, χ2 = 20.05, p < 0.001), and medRxiv to Research Square (df = 1; χ2 = 37.36; p < 0.001), however, we did not find a significant difference when comparing bioRxiv to medRxiv (df = 1; χ2 = 1.53; p = 0.215).

In our sample, the median time interval from posting on a preprint server to publication in a scholarly journal was 24 days (minimum: 0; maximum 117). BioRxiv had the mean shortest time interval (26.0 days; median: 20; Q1–Q3: 13–38.8), followed by medRxiv (mean: 27.7 days; median: 23; Q1–Q3: 11–42.5) and Research Square (mean: 65.7 days; median: 55.5; Q1–Q3: 74–85.8; p < 0.001) (Fig. 2). In a post-hoc Dunn’s test we identified a significant difference between bioRxiv and Research Square (z = −113.77; p-adj < 0.001), and, between medRxiv and Research Square (z = −111.11; p-adj < 0.001), however, we did not find a significant difference between bioRxiv and medRxiv (z = −2.66; p-adj = 1.000).

Figure 2 Time interval from posting on preprint server to publication in a scholarly journal.

BioRxiv preprints had the highest mean number of citations (5.50; median: 2; Q1–Q3: 0–6), followed by medRxiv preprints (mean: 2.06; median: 0; Q1–Q3: 0–1) and Research Square (mean: 0.23; median: 0; Q1–Q3: 0–0; p < 0.001) (Table 3). In posthoc pairwise comparisons, we identified significant differences between all pairs of preprint servers in terms of the number of citations (all p-adj < 0.001).

Table 3 Citation counts for preprints posted in medRxiv, bioRxiv and Research Square.

	medRxiv (n = 225)	BioRxiv (n = 45)	Research Square (n = 18)	p-value	
Mean	2.06	5.50	0.23	p < 0.001	
Range	170	168	16		
Minimum	0	0	0		
Maximum	170	168	16		
Median (Q1–Q3)	0 (0–1)	2 (0–6)	0 (0–0)		
Note:

Q1, quartile 1; Q3, quartile 3.

Discussion

Preprints intend to accelerate the access to preliminary data for the scientific community, mainly to rapidly inform about time-sensitive issues and receive rapid feedback before entering a peer-review process, which is a requirement for publication in the majority of indexed journals (Mudrak, 2020). medRxiv and bioRxiv, two of the most widely used preprint servers, have a disclaimer on their homepage that states: “these are preliminary reports that have not been peer-reviewed. They should not be regarded as conclusive, guide clinical practice/health-related behavior, or be reported in news media as established information” (medRxiv, 2020).

Scientific production has drastically increased due to the COVID-19 pandemic, and approximately 900 articles, including published papers and preprints, were available before March 12, 2020 (Callaway et al., 2020). During the pandemic, multiple scholarly journals required authors to share their papers simultaneously using a preprint server (Fidahic et al., 2020).

We found that the publication rate for preprints was very low (5.7%), even after a considerably long follow-up to determine the publication status. Palayew et al. (2020) described that on average, 367 articles related to COVID-19 were published weekly in scholarly journals and that the median time from submission to acceptance was just 6 days, which is remarkably shorter than the usual time to accept that scholarly journals had before the pandemic. Thus, we consider that enough follow-up time was provided to identify most of the preprints in our sample that will be published in a journal.

Our findings reveal a drastically lower publication rate for preprints related to COVID-19 when compared to the publication rates of preprints related to other public health emergencies of international concern (PHEIC). For example, during the Zika outbreak (November 2015 to August 2017), 174 preprints were posted, with a publication rate of 60%; while 75 preprints were posted during the Ebola outbreak (May 2014 to January 2016), with a publication rate of 48% (Johansson et al., 2018). During the COVID-19 pandemic, the number of preprints was unprecedented, and a potential saturation of scholarly journals might contribute to this lower publication rate.

However, on a rapid search on the iSearch COVID-19 portfolio, we identified that 17,564 articles had been published in peer-reviewed journals indexed in PubMed during the same interval of time (January 31, 2020 to May 31, 2020). These findings suggest that despite being swamped with papers by the scientific community from all around the globe, scholarly journals have been able to maintain a substantial output of peer-reviewed publications, and we could assume that other factors, such as overall lower quality, might contribute to the low publication rate for preprints in our sample.

Never before in history has the scientific community produced so much non-peer-reviewed data (Dinis, 2020). However, by attempting to drastically increase the output of scientific data during times of desperate need for information about a novel disease, we might overlook essential processes needed to promote the quality and validity of this scientific production. In this report, we found that articles published in a scholarly journal had a significantly higher citation count than preprints, and even though we did not directly evaluate the quality of the preprints, the citation count is an indicator of the scientific impact, which is one of the components within the concept of scientific quality (Aksnes, Langfeldt & Wouters, 2019). Importantly, a significant proportion of preprints in our sample were never cited by other scientific papers, which highlights the low scientific impact that some preprints had, and probably did not merit a fast, non-peer-reviewed publication. As well, we could infer that articles published in scholarly journals are used more often by the scientific community to develop new papers, perhaps due to higher quality and the confidence that a peer-reviewing process might provide. These findings are supported by the fact that amongst preprints that were eventually published, the citation counts once published in a scholarly journal were significantly higher than for the preprint version However, these might also reflect longer times in article form compared to preprint form or growing awareness of the research over time.

In a study conducted before the pandemic, there were small differences in quality between preprint versions and articles published in scholarly journals (Klein et al., 2019). However, these findings are not necessarily applicable to the current situation, as preprint production has increased drastically due to the COVID-19 pandemic, and our results support the impression of a lower scientific impact of preprints when compared to scholarly journals.

Our study also revealed differences among the three preprint servers analyzed. There, medRxiv had a higher publication rate, while bioRxiv had the shortest interval from posting to publication and had the highest citation counts. These differences could be related to the scope that preprint servers have and how visible these preprint servers are for the scientific community, which could determine the scientific impact that a certain publication will have.

Some preprints might contain essential and time-sensitive information, for example, the basic reproduction number (R0), calculated using data on preprints was shown not to be different from the one estimated in peer-reviewed articles (Majumder & Mandl, 2020). Similarly, preprints on the viral sequence and structure have allowed for the early investigation of potential therapeutic options and vaccines (Brainard, 2020; Kwon, 2020). While there is widespread agreement that preprints could be useful in the current context, there are significant risks associated with the potential spread of faulty data without appropriate third-party screening (Dinis, 2020).

The basic screening process employed by preprint servers may not be enough to avoid the dissemination of flawed information (Rawlinson & Bloom, 2019). For example, a preprint that was posted on bioRxiv suggested significant molecular similarities between SARS-CoV-2 and HIV (Kwon, 2020). This preprint was later withdrawn from the server, however, by the time that happened, it had already sparked controversy and conspiracy theories. To our concern, we found that during the COVID-19 pandemic multiple preprints have been used in the development of clinical guidelines and public health policies (Bhimraj et al., 2020; Nicolalde et al., 2020).

Although peer-review aims to be an exhaustive and in-depth process that improves the quality of a manuscript, articles published in a peer-reviewed journal should not be taken as non-refutable knowledge. To illustrate this, a couple of high-impact peer-reviewed articles were withdrawn from two prestigious journals due to significant concerns on primary data validity (Mehra, Ruschitzka & Patel, 2020; Mehra et al., 2020). Currently, 33 papers related to SARS-CoV-2 and COVID-19 (13 preprints and 20 articles published in scholarly journals) have been retracted (Retraction Watch, 2020). It is possible that due to the fast-track processes that many scientific journals currently provide, the quality of peer-reviewed articles might also be compromised (Palayew et al., 2020).

We acknowledge certain limitations to our study. Even though we had an extensive sample size, and included the most used preprint servers, several other platforms host preprints. As well, some preprints included in our sample might end up published in a scholarly journal after some time, however, considering the shortened editorial processes due to the COVID-19 pandemic (Palayew et al., 2020), we consider that we provided a sufficient follow-up period to identify most of the preprints that will be eventually published within our sample. Additionally, our analysis approach for citation counts did not permit adjusting for other variables, which could contribute to confounding bias. We consider that a potential confounding factor is the time interval from posting on a preprint server, as preprints that were available for a longer time online might have a higher number of citations than more recent preprints. We consider that by using comprehensive platforms, such as the iSearch COVID-19 portfolio, and resources that include verified data, such as Crossref, our results are reliable.

Conclusions

Preprints have been used widely as a means to share information rapidly during the COVID-19 pandemic, and multiple organizations and scholarly journals have urged authors to share their research on preprint servers. However, the publication rate for preprints in our sample was remarkably low, despite accelerated editing processes by multiple scholarly journals. There are some important caveats related to the use of preprints. Notably, it should be highlighted that preprints include non-peer-reviewed data, and thus, should be considered as preliminary reports. We found that preprints that had sufficient quality to be published after going through peer-reviewing scrutiny were more cited than preprints that remained unpublished. In this sense, preprint servers might provide an opportunity for higher visibility and potentially improve the scientific impact of research articles. Peer-reviewing and editing processes play an essential role during the current infodemic, and researchers that initially post their preliminary reports on preprint servers should strive to publish their works in peer-reviewed journals. Also, researchers should be careful when using and citing preprints in their publications.

Supplemental Information

Supplemental Information 1 Raw data of preprints posted between Jan to May 2020.

Click here for additional data file.

Additional Information and Declarations

Competing Interests

Author Contributions

Data Availability

The authors declare that they have no competing interests.

Diego Añazco conceived and designed the experiments, performed the experiments, analyzed the data, prepared figures and/or tables, authored or reviewed drafts of the paper, and approved the final draft.

Bryan Nicolalde conceived and designed the experiments, performed the experiments, analyzed the data, prepared figures and/or tables, authored or reviewed drafts of the paper, and approved the final draft.

Isabel Espinosa conceived and designed the experiments, performed the experiments, analyzed the data, prepared figures and/or tables, authored or reviewed drafts of the paper, and approved the final draft.

Jose Camacho conceived and designed the experiments, performed the experiments, analyzed the data, authored or reviewed drafts of the paper, and approved the final draft.

Mariam Mushtaq conceived and designed the experiments, performed the experiments, analyzed the data, prepared figures and/or tables, authored or reviewed drafts of the paper, and approved the final draft.

Jimena Gimenez conceived and designed the experiments, performed the experiments, analyzed the data, prepared figures and/or tables, authored or reviewed drafts of the paper, and approved the final draft.

Enrique Teran conceived and designed the experiments, analyzed the data, authored or reviewed drafts of the paper, and approved the final draft.

The following information was supplied regarding data availability:

Raw data are available as a Supplemental File.

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
