# Peer review of "Publication rate and citation counts for preprints released during the COVID-19 pandemic: the good, the bad and the ugly"

_PeerJ, doi:10.7717/peerj.10927_

## Round 0.1 · original submission · Minor Revisions

Thank you for submitting this new version of your manuscript. I think you have done well in addressing the issues raised. One of the reviewers (#2) was happy with the revisions from your previously submitted version and had no further comments. The other reviewer was positive about your work but has made a number of comments that need attention. I will ask you to respond to each of their comments with an explanation of what changes you have made in response to that particular comment or an explanation of why you have not made any such changes. I’ll ask you to do the same for my comments below.

1. Related to a comment from Reviewer #1, please note that non-parametric approaches are not justified by “data ha[ving] a non-normal distribution” (Line 113). The assumptions behind linear models, including t-tests, ANOVAs, and ANCOVAs, as well as more generally, concern model residuals as indicating some properties of the error term in the population. For models with only factor variables, examining the model residuals is related to examining the data within each combination of factors, but the assumptions are specifically around the error term and if explored through the data, this is done through looking at the model residuals.

2. Kruskal-Wallis should be accompanied by post-hoc Dunn’s tests (Line 117). I am assuming the p-value on Lines 162 and 165 are from KW tests and readers will appreciate you also providing the pairwise comparison p-values around there. A similar point applies to the Chi-squared test on Line 159 where the following text would be more informative with relevant pairwise p-values from 2-by-2 Chi-squared tests.

3. The alpha on Line 120 is presumably two-sided (except for the Chi-squared tests).

4. Technically the range is the difference between the minimum and maximum (https://en.wikipedia.org/wiki/Range_(statistics)) and what is shown on Line 128 is the minimum–maximum values. The same point arises elsewhere and also applies to the IQR (https://en.wikipedia.org/wiki/Interquartile_range) on Line 135 and elsewhere (presenting the 25th and 75th percentiles is fine, and more informative than the IQR, it’s just making sure that the values are described as such). Table 1 appears to show ranges as defined by statisticians (but not when it comes to IQRs).

5. While your analysis approach doesn’t permit looking at adjusting for other variables, this makes comparisons such as on Line 144–147 potentially vulnerable to confounding. It seems possible, but obviously not certain, that preprints that have been published or cited more often had been available longer or were initially uploaded to different preprint servers. You could either use another form of regression (logistic or Poisson [with robust standard errors] regression might be worth looking at for publication and negative binomial regression might be worth thinking about for citations, where quantile regression would be another option) so that you can try to reduce the risk of confounding bias, or you need to acknowledge this limitation more clearly (perhaps around Lines 187 [note also the claim on Line 185 that “most” will have been published could potentially be justified using the 90th or 95th percentiles for time to publication for preprints described on Lines 190–193 if this is available], 219, 229, and/or 254–261). As another example of the risk of confounding bias, it seems at least possible that the different preprint servers were used more at different points in time (were they all as well-known at the start? Are there differences in perception?)

6. There are some small parts of the results (Lines 147–149 and “Remarkably” on Line 150) that I feel would be better in the discussion. Try to avoid interpreting or judging results in the results section.

As Reviewer #1 notes, there is work needed on language. I’ve made some suggestions below for the abstract as illustrations, but you will need to address the writing in the manuscript as a whole. One option would be to use an editing service, such as the one provided by PeerJ.

Line 21: “In December…” or “On December XXth…” if you have a specific date in mind.

Line 24: “…in scholarly journals…” (See Reviewer #1’s comments also.)

Lines 27–28: Perhaps “…obtain metadata using the Crossref public API…”

Line 28: Rather than “different analyses”, perhaps “comparisons”. Better still would be to identify (briefly) the nature of these comparisons in the abstract.

Line 30: “…publication in a scholarly…”

Line 32: “…published in scholarly…”

Line 33: While not a language issue, consider whether the second decimal place in “5.69%” is meaningful or whether “5.7%” is how you might describe this result to a colleague.

Line 34: “…published in scholarly…”

Lines 35, 36, and 38: While not a language issue, note that very small p-values are sensitive to small changes in the data (potentially changing by one or several orders of magnitude with one or a small number of changes) and to violations of the mathematical assumptions behind the test. For this manuscript, I suggest using “p<0.001” for all p-values smaller than 0.001, as is conventional outside of omics research and similar. Note also that decimal points (“.”) should be used consistently rather than commas for p-values (c.f. Line 36 and the percentage on Line 33).

Line 37: “…publication in a scholarly…”

Line 40: “…publication in a scholarly…”

Line 47: “…published in a peer-reviewed…”

The same, similar, and other language issues can be identified in the body of the manuscript by careful proofreading.

Reviewer 1 ·

Basic reporting

A well written original article. The topic is attractive and the results are useful. Aime and the scope of the article are correct.
The text should be written in clear English language, by native speaker or professional translator.
I suggest to place a specific timeline figure where would be described the period of the search, and also the date of search. The title should be more comprehensive, discussion should be shorter and the conclusion should be longer, without any specific and opaque words

Experimental design

Is completely correct with minor suggestions. Methodology is also correct.

Validity of the findings

Is completely correct and in line with journal policy and suggestions with minor suggestions.

Additional comments

A well written original article. The topic is atractive and the results are useful. Aime and the scope of the article are correct.
The text should be written in clear English language, by native speaker or profesional translator.
I suggest to place a specific timeline figure where would be described the period of the search, and also the date of search. The title should be more comprehensive, discusion should be shorter and the conclusion should be longer, without any specific and opaque words

Title:
Line 2: I find the part of the title after „:“ unnecessary (The good, the bad and the ugly), so I suggest to remove it.
Abstract is good structured.
In line 24, and also in the all other text is written: „published on scholarly journals“, but I find more in line with native English language „published in scholarly journals“ to be written. (it should be considered by editor).
Line 28: Te sentence start: „We then determined“. I suggest to be written: „After that we determined“. I find it more in line with native English.
Introduction: in line 52 after the word „report“ should add „of scientific work, “ (full sentence should be: A preprint is a preliminary report of scientific work, that is shared publicly before it has been peer-reviewed.
In line 58 „has been controversial“ should be replaced with „is being disscused“.
In line 77 at the end should delete „that“
In line 82 „works“ should be replaced with „publication papers“ and also afterward in the text.
In „Statistical analysis“ you should describe why you have chosen specific tests for analysis, so it can be clearly understood.
Results: in line 127 -128 is written data of median time from the date of posting to the date of search!? It is more important to publish the median time of posting to the date of publication in a scholarly journals.

In line 132 sentence that with Table 1 is unnecessary because at the end of the line 137 it is correctly declared.
In line 148 should escape word „eventually“ or the results to be more better described.

In line 165 „)“ is missing. Also (Table 3) need to be in separate bracket.

Discussion is too long. It should be shorten.

At the end of the line 174 reference is needed.

Sentence 177-179 is crutial but too long and opaque. It should be reformulated.
Sentence 184 – 187 is too long, and opaque. It should be more in line with english language (even 4x you hve word „that“ in the sentence)

Sentence 193 – 195 is opaque. It should be more understandable and reformulated.

The state in line 216 and 219 need reference if it is from other papers. In case that it is from your work it should be in section „Result“.

Conclusion: It should be longer. The last sentence (line 268 – 269) is controversal, and should be deleted. Also in conclusion should be more stated to justify or not the use of the preprint servers in future and to publish or not IN PREPRINT SERVERS in line with your results.

Table 1. should be added on the left side with column in which will be explained data in the table
Table 2. should be added on the left side with column in which will be explained data in the table

Also should be considered to add separate analysis for each preprint server (medRxiv, bioRxiv and RS) in Table 1 and Table 2.

Annotated reviews are not available for download in order to protect the identity of reviewers who chose to remain anonymous.

Reviewer 2 ·

Basic reporting

Adequate

Experimental design

Adequate

Validity of the findings

Adequate

Additional comments

The authors have revised the manuscript adequately

---

## Round 0.2 · Minor Revisions

Thank you very much for your constructive revisions and thoughtful responses to the reviewer’s and my questions. I have one remaining query of some (potential) substance and a number of minor, mostly wording, queries and suggestions that it would be best to address now rather than later. I don’t believe that these will be difficult for you to address/rebut and subject to this and no other issues arising, I would anticipate being able to quickly accept your revised manuscript.

Looking again at Table 1 and reproducing the values there based the data you’ve kindly provided, it struck me that the first comparison (unpublished pre-prints [mean 1.99] to published pre-prints [mean 65.8]) might make more sense by including both citations as pre-prints and as articles for the latter (being their total number of citations). I believe that this would make the means 1.99 versus 65.8+6.29=72.1 (with values 0–3757, median 9 and 25th and 75th percentiles 2 and 38). Otherwise, the comparison seems slightly odd to me, being two different types of articles (unpublished and published) and two different venues (preprint serve and journal) and so meaning that two things are varying rather than a more interpretable one thing, but perhaps I’m not quite following what you’re intending to compare here. I don’t think this would result in any meaningful changes, just updating a few values in the text and tables. If you want to argue for the current presentation, I’m very happy to hear your thinking there.

Line 28: I’m not sure that “different” (in “and made different comparisons”) is needed, or in fact, what it’s telling the reader. If you agree, it could be deleted.

Line 33: Similarly, I’m not sure that “also” (in “We also found that”) is needed here since at this stage only the publication rate has been presented and this sentence is the first inferential result presented in the abstract.

Lines 34: I think this might be clearer as “…significantly higher citation count than UNPUBLISHED preprints…” (as you use on Line 36) as the comparison in Table 1 is with those 4765. If you agree with my more substantial comment above, this could be “…significantly higher TOTAL citation count than UNPUBLISHED preprints…” to emphasise the total being used here. This might also require edits to Lines 141–144 (e.g., “UNPUBLISHED preprints (n=4765)…while…had a TOTAL mean citation count of 72.1… After conducting…had a significantly higher TOTAL number of citations than UNPUBLISHED preprints...”)

Lines 37–38: It might help the reader if this comparison was clarified, e.g. “…after publication in a scholarly journal COMPARED TO AS A PREPRINT (p<0.001).”

Line 66: This is the only instance of a blank line between paragraphs I can see.

Line 67: Simply as a comment, “Coronavirus disease 2019” returns ~13 million hits in Google, whereas “Coronavirus disease 19” returns a relatively modest (!) 276 thousand. Having said that, some sites, including the CDC’s, appear to use both forms, although the latter is often hyphenated.

Line 68: “…on March 12…” seems more usual than “…by March 12…”.

Line 78: Perhaps “…initially ON a preprint server…”, or “using” (rather than “in”). The same applies to Line 195 and similar to Line 281.

Line 94: Sorry for not asking about this earlier, but how were the 45 duplicates handled in terms of the descriptives and analyses (if a preprint was uploaded to two or all three of the servers)? Were the duplicates all within-server, which would perhaps resolve this question quickly (but should be mentioned in the manuscript if so)? If not, was the publication status and the number of citations combined across the multiple servers (e.g. if one uploaded document was cited 10 times and the same document on another server with a different DOI was cited 10 times, did this count as 20 citations for the preprint)? Also, in the Chi-squared and Kruskal-Wallis analyses (Lines 122–125), did you take the first uploaded server for the purposes of comparing servers?

Line 120: As well as the MW-U tests for the independent samples, the paired analysis (before and after publication for those 283 preprints, results on Lines 148–149) would have been done using a Wilcoxon signed-ranked test I’m guessing.

Line 141: The comma in “1,99” should be a period (“1.99”).

Line 149: This p-value is, I’m guessing, from a signed Wilcoxon test? Just to check, using Stata, I get a test statistic of z = -12.43 here for these n=283.

Line 177: I suggest still using three decimal places for this p-value for consistency, i.e. “p-adj=1.000”.

Line 178: I suggest using two decimal places here (i.e. “5.50”) so the reader doesn’t wonder if a digit was omitted when seeing the two decimal places for the other servers.

Line 181: Perhaps “…between all PAIRS OF preprint servers…”, or “combinations of” to emphasise the pairwise nature of these tests. As well, or alternatively, Line 182 could read “(all p-adj<0.001)” to emphasise that there were multiple tests involved here.

Line 186: I wonder if “…rapidly inform ABOUT time-sensitive issues…” might read more naturally.

Line 201: Perhaps “…consider THAT enough follow-up TIME WAS PROVIDED to identify…”

Line 208: Perhaps “During THE COVID-19 pandemic…”

Line 220: I think “guarantee” is a very strong word to use here, particularly given some topical and some non-topical but very high-profile retractions after problematic research got through the peer-review and editorial processes. Perhaps “enhance”, “promote”, “check”, or “gatekeep”? C.f. with your Lines 256–260.

Line 229: Again, “guarantees” is a very strong word (even with the “might”). Perhaps “reassurance” or “confidence”?

Lines 229–231: Wouldn’t some of the published preprints, particularly those of higher potential impact (early in the pandemic) and/or quality (and so published faster), have had more time in the study period as publications than as preprints? (E.g., at the extreme, a preprint uploaded in January that was published two weeks later could have several months to accumulate citations as an article.) Citations per day (or week or month or annualised…) would normalise for this, but more time and citations will also lead to more awareness of the work and, for my own publications, citation numbers often suggest a snowball-like effect as awareness among research groups grows, before reaching saturation, eventually starting to drop off as newer research takes priority. (c.f. with your Lines 272–274) If you agreed here, you could add (on Line 231), something like: “However, these might also reflect longer times in article form compared to preprint form or growing awareness of the research over time.” This is merely a thought that occurred to me while re-reading your manuscript and you should feel free to reject the suggestion.

Line 235: Perhaps “showed” is an overly strong word here. Maybe “suggest” or “support the impression of”?

Line 237: Perhaps “There” rather than “Then”.

Line 243: “…preprints WAS showN not…”

Line 244: Perhaps “…2020). SIMILARLY, preprints…”

Line 246: I’d delete the “a” from “While there is a widespread agreement”.

Line 263: Perhaps “provide” rather than “handle”? Or “offer”.

Line 266: “…included THE most used preprint servers…”

Line 285: Do you mean “We found that preprints that had sufficient quality to BE PUBLISHED AFTER goING through peer-reviewing scrutiny…”? I’d delete the “commonly” on Line 286 so it reads “…were more cited than…”

Line 290: “…works IN peer-reviewed journals…”

Line 315: Could you replace this URL with a shortened doi.org one?

---

## Round 0.3 · Minor Revisions

Thank you very much for the revised version and your rebuttal. While I didn’t make the point explicitly, and my apologies for that, I was anticipating that any changes you made to Table 1 (here the update to using total citations for the first comparison) would also be made to Table 2, which repeats the analyses after excluding those without citations (a definition that I think needs some clarification, see point 1 below). I can replicate the values in Table 1 and feel that I understand the logic there, so my comments here are only about Table 2 and these secondary/sensitivity analyses. I don’t think that any changes in response to my questions will have any effect on your conclusions.

1) First, it seems to me that to produce the n=174 shown in some parts of the second results table, it is necessary to exclude all eventually published preprints with no citations as preprints (n=93, as you note on Line 165), and then to also exclude those with no citations as articles (n=50 in total, with n=16 being additional exclusions, giving n=109 distinct articles having no citations in one of the two forms). Excluding only the former n=93 (as described on Line 165) seems to result in n=1718 and n=190 (not n=174), a frequency also seen in this table (in the second and third comparisons). I wonder if you could explain your final approach for dropping those with no citations in the manuscript around Lines 166–168 so the reader knows exactly what they are looking at in Table 2. The second comparison here looks at the n=190 versus the n=174, but as this is a paired analysis, the test is only using the n=174 with data for both (the other n=16 were completely excluded from the analysis), so it seems to me that the current descriptives provided are not quite consistent with the test. Should the n=190 for the second and third comparisons be replaced with the n=174, or should the n=174 for the first and second comparisons both be n=190? Note that in Table 1, the published articles are n=283 for all four instances in the table, and the same consistency seems to me to be appropriate for Table 2 also, unless of course there is a particular reason for having different n’s for this set of published articles that were cited depending on the comparison being made. Since Table 2 drops those preprints that were not published as articles with no citations as preprints, it seems to me that the most natural option would be to do the same for those preprints that were published as articles, i.e. use the n=190 option throughout the table here.

2) Either way, I’m pleased that you have used the total publications as the first comparison in Table 1. As noted above, Table 2 should be consistent with this approach. Based on my understanding of the data and using n=174, this would result in a median of 24 (rather than 20.5) and a mean of 114.67, with changes to some of the other values also. For n=190, these values would be 22 and 105.14 respectively. Please do let me know if I’m misunderstanding your data and/or the calculations here.

Please carefully check all numbers in the table and text before you submit your revised manuscript so that I can quickly accept this version.

I’ll also make a few small wording suggestions for newly added/edited text to minimise any editing of the manuscript later on.

Lines 96–97: “For duplicated preprints, we decided to USE the version that was first uploaded TO a preprint server.”

Line 125: “before and after publication IN a scholarly journal”

Line 155: “the same set OF articles”

Well done on writing what I think will be a useful addition to the literature on an interesting topic. I look forward to seeing the final version of the manuscript addressing the above.

---

## Round 0.4 · accepted · Accept

Thank you very much for your responses and revisions. I am very happy to accept your manuscript and I look forward to seeing it “in print” in the near future. I will note a couple of very minor edits to new/newly revised text and a couple from older text that you should address in preparing the final version of your manuscript.

Line 97: “use of the version” should be “use the version” or similar (deleting “the”).

Lines 169–170: “However, from the latest 34 were already included in the previous list, then only an additional 16 were removed for a final sample of 174 published preprints with at least one cited.” is a little unclear and somewhat awkward. This could be worded as “However, 34 of these 50 were already included in the 93 with no citations as preprints, and so only an additional 16 were removed for a final sample of 174 published preprints with at least one citation.” or similar.

Lines 181–182: “a preprint server to publication on a scholarly journal” should be “a preprint server to publication in a scholarly journal” or similar (“on” becoming “in”).

Line 183: For consistency in decimal places with the other means (27.7 and 65.7), “26” should be “26.0” so the reader doesn’t wonder if a digit is missing.